

**Emission Factors and Optical Properties of Black and Brown Carbon Emitted at a Mixed-**
**Conifer Forest Prescribed Burn**
James D.A. Butler,[1,2] Afsara Tasnia,[3] Deep Sengupta,[4] Nathan Kreisberg,[5] Kelley C. Barsanti,[3,6]
Allen H. Goldstein,[1,5] Chelsea V. Preble,[1,2] Rebecca A. Sugrue,[1,2] Thomas W. Kirchstetter[1,2,*]
[1]Department of Civil and Environmental Engineering, University of California, Berkeley,
Berkeley, California 94720
[2]Energy Technologies Area, Lawrence Berkeley National Laboratory, Berkeley, California 94720
[3]Department of Chemical and Environmental Engineering, Center for Environmental Research
and Technology, University of California, Riverside, Riverside, California 92507
[4]Department of Environmental Science, Policy, and Management, University of California,
Berkeley, Berkeley, California 94720
[5]Aerosol Dynamics Inc., Berkeley, California 94720
[6] Atmospheric Chemistry Observations and Modeling, U.S. National Science Foundation
National Center for Atmospheric Research, Boulder, Colorado 80301
*Corresponding author email: twkirchstetter@lbl.gov



**Abstract**

Prescribed burning is a fuel management practice employed globally that emits carbonaceous aerosols that affect human health and perturb the global climate system. Aerosol black and brown carbon (BC and BrC) emission factors were calculated from ground and aloft smoke during prescribed burns at a mixed-conifer, montane forest site in the Sierra Nevada in California. BC emission factors were $0.52 \pm 0.42$ and $1.0 \pm 0.48$ g kg$^{-1}$ for the smoldering and flaming combustion phases. MCE is a poor predictor of BC emission factor, in this study and published literature. We discuss limitations of using BC to PM$_{2.5}$ mass emission ratios to generate emissions inventories. Using BC emission factors measured in this study, we recommend BC to PM$_{2.5}$ ratios of 0.7% and 9.5% for the smoldering and flaming combustion. We calculated absorption Ångström exponents (AAE) based on multiwavelength absorption for BrC and BC of 6.26 and 0.67. Using the Delta-C method with a BrC-specific absorption cross-section, we estimate a smoldering combustion BrC emission factor of $7.0 \pm 2.7$ g kg$^{-1}$, nearly 14 and 7 times greater than the smoldering and flaming BC emission factors. Furthermore, we estimate that BrC would account for 23% and 82%, respectively, of the solar radiation absorbed by the smoldering smoke in the atmosphere integrated over the solar spectrum (300–2500 nm) and in the UV spectrum (300–400 nm), indicating that BrC affects tropospheric photochemistry in addition to atmospheric warming.

**Key Figure**

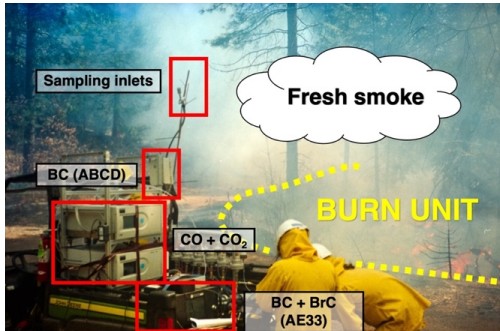



**Keywords**

Prescribed burn, black carbon, brown carbon, emission factor, light absorbing aerosol, wildland

fire fuel consumption model

**1 Introduction**

Prescribed burns are controlled burns that consume excess and dead fuel in an ecosystem,

like the duff, shrubs and dead biomass in the forest understory, or floor. In contrast, wildfires are

uncontrolled burns that may consume both the understory and overstory, or canopy, of a forest

and may spread to nearby property, endangering the homes and lives of people in the wildland

urban interface. Routine prescribed burns, or other fuel management practices like mechanical

thinning, reduce the risk and severity of wildfire ignition in forest ecosystems and increase the

resistance to ecosystem transition (i.e., conversion of forest to shrubland) caused by wildfires

(Steel et al., 2021; Wu et al., 2023).

Prescribed burns and wildfires emit fine particulate matter ($PM_{2.5}$), carbon monoxide

(CO), carbon dioxide ($CO_2$), volatile organic compounds, and nitrogen oxides (Andreae, 2019;

Urbanski, 2014; Urbanski et al., 2008; Wiedinmyer et al., 2006). Emitted $PM_{2.5}$ includes organic

aerosol, some of which is light-absorbing brown carbon (BrC), and black carbon (BC). Whereas

BC absorbs solar radiation broadly across the visible spectrum, BrC light absorption is highly

wavelength dependent and strongest in the near-UV spectral region (Bond et al., 2004;

Kirchstetter et al., 2004; Laskin et al., 2015). Due to their perturbation of the radiative balance of

the atmosphere and short atmospheric residence time compared to $CO_2$, BC, and BrC are

considered short-lived climate forcers (Feng et al., 2013; Kirchstetter and Thatcher, 2012; Zhang

et al., 2020). Additionally, BC and BrC surface deposition in snowy climates reduces the solar

reflectance of snow and may accelerate snow melt (Chelluboyina et al., 2024; Hadley and

Kirchstetter, 2012; Kaspari et al., 2015; Yang et al., 2015). Human exposure to carbonaceous

aerosols also has detrimental health effects including cardiovascular disease, lung cancer,

adverse birth outcomes, and premature mortality (Dong et al., 2023; Grahame et al., 2014;

Janssen et al., 2011). Wildland fires are a major source of pollution relevant to human exposure

and account for one third of total $PM_{2.5}$ emissions in the U.S., with roughly equal contributions

from prescribed burns and wildfires (Larkin et al., 2020).



Wildland fire modeling frameworks, or smoke models, estimate the amount of smoke
emitted during burn events to create input emissions necessary for climate modeling, air
pollution modeling, and health impact assessments (California Air Resources Board, 2020;
Connolly et al., 2024; Cruz Núñez et al., 2014; Maji et al., 2024). Smoke emissions from
wildland fires are estimated with fuel consumption models like Burnup, part of the First-Order
Fire Effects Model (FOFEM), and CONSUME, part of the BlueSky Smoke Modeling
Framework (Keane and Lutes, 2018; BlueSky Modeling Framework, 2024). Both smoke models
compute total emissions of a pollutant by multiplying pollutant emission factors by the mass of
fuel consumed during both the high intensity and low intensity stages of a burn event, which
roughly correspond to the flaming and smoldering phases of a wildland fire.
The differences in fuel mass consumption and temperature in these phases affect the
emission rate of pollutants, sometimes by an order of magnitude. In the flaming phase, fuel mass
consumption and temperature are highest and combustion is more complete, while both are lower
in the smoldering phase that is characterized by incomplete combustion (Urbanski, 2014).
Flaming combustion generally has a higher emission rate of BC and a lower emission rate of BrC
compared with smoldering combustion, while smoldering combustion is marked by higher
emissions of CO and BrC (Chen et al., 2007). Experiments designed to quantify pollutant
emissions must consider the placement of sampling instrumentation to capture these distinct
combustion phases of a burn, with aerial sampling platforms more likely to capture a mixture of
flaming and smoldering combustion due to the convective lofting of smoke caused by flaming
combustion (Aurell et al., 2021). Ground-level smoke, on the other hand, tends to be dominated
from smoldering combustion (Aurell and Gullett, 2013).
In this study, we conducted field sampling of pollutant emissions from prescribed burning
of a mixed-conifer understory and computed BC emission factors, BrC emission factors, and
aerosol absorption properties with ground and aerial sampling platforms. We investigate the
relationship between BC emission factors and combustion conditions and, finding that the
modified combustion efficiency (MCE) is a poor predictor of BC emission factor, propose a
framework to aggregate emission factors by either flaming or smoldering conditions to convey
the average value and variability of emission factors within these combustion regimes in fuel
consumption models. We report $BC/PM_{2.5}$ ratios, or speciation profiles, for a mixed-conifer





understory prescribed burn. We then discuss how applying an incorrect BC/PM$_{2.5}$ ratio in
wildland fire modeling frameworks may lead to large errors in BC emissions, using the
ecosystem studied in this work as an example. We compute the absorption Ångström exponent
(AAE) for the prescribed burn smoke aerosols, demonstrating that AAE is sensitive to the
wavelengths used in its calculation, and present estimates of AAE separately for BC and BrC to
estimate their contributions the solar radiation absorbed by the smoldering smoke in the
atmosphere.

**2 Materials and Methods**
**2.1 Field Measurements**

Field measurements were made at the Blodgett Forest Research Station (38.915224, -

120.662420), located 1370 meters above sea level on the western slope of the Sierra Nevada, 21
km east of Georgetown, CA. Prescribed burns were conducted in a mixed conifer forest, typical
of montane ecosystems of the Sierra Nevada (North et al., 2016). Three forest units were burned
in consecutive days in April 2021, as shown in Figure S1 in the Supporting Information (SI). The
prescribed burn on the first day escaped the designated unit (A) and the burn was terminated
early. The remainder of unit A was burned on the second day and units B and C were burned on
the third and fourth days, respectively.

Prescribed burn smoke was measured using both a ground and an aerial sampling

platform. The ground platform consisted of pollutant analyzers secured to a utility task vehicle
stationed immediately downwind of the fire to capture fresh smoke two meters above ground
level (see Figure S2). The ground platform was moved once each day as the burns progressed
and winds shifted to be on service roads nearby the highest intensity burn activity and the aerial
platform takeoff/landing location. Across the four days, there were nine ground sampling
sessions: at two locations on each day, plus one "next-day" smoldering sample on the second day
for burn unit A before ignition of the remaining unit. For the aerial platform, pollutant analyzers
were attached to the unmanned aerial vehicle, a DJI Matrice 600 Pro hexacopter. Concurrent
with ground sampling, the unmanned aerial vehicle was flown 23 times across the four days with
at least five flights each day and a maximum of 10 flights on the third day. The average elevation



of aerial platform throughout sampling was 29 meters, with an average sampling elevation range
of 16–42 meters across all flights.

BC, CO, and $CO_2$ were measured on both the ground and aerial sampling platforms. BC

was measured using two filter-based aerosol absorption photometers: the Aerosol Magee
Scientific aethalometer model AE33 with a 2.5 μm cyclone on the inlet on the ground platform
and the custom-built Aerosol Black Carbon Detector (ABCD) on both the ground and aerial
platforms (Caubel et al., 2018; Sugrue et al., 2024). The ABCD estimates BC concentration
based on aerosol optical attenuation at 880 nm wavelength (λ). The AE33 also measures BC at
λ=880 nm, in addition to aerosol optical attenuation at six other wavelengths. In particular, the
AE33 reports the mass concentration UV-absorbing aerosol (UVPM) based on the optical
attenuation at 370 nm. BrC concentration was estimated from these data as described below in
Section 2.2. Collocating the AE33 with the ABCD on the ground enabled an analysis to express
BC measured with the ABCD in terms of AE33 equivalence, also described below. CO and $CO_2$
were measured by non-dispersive infrared (NDIR) absorption photometry on the ground
platform using Horiba models APMA370 and APCA370, respectively.(Tasnia et al., n.d.) CO and
$CO_2$ were measured on the aerial platform with an electrochemical cell (Alphasense CO-B4) and
NDIR sensor (PP Systems SBA-5), respectively. All instruments reported pollutant
concentrations at 1 Hz frequency. Data were post-processed and validated prior to analysis using
the quality assurance and control measures described in the SI, including pollutant concentration
time-series alignment and loading artifact correction of BC concentrations measured with the
ABCD.

**2.2 Calculations**

Light absorption by carbonaceous aerosols increases with decreasing wavelength, a trend

that is often modeled as a power-law:
$$b_{abs}(\lambda) \propto \lambda^{-\text{AAE}}$$                         (Equation 1)
AAE was calculated by an ordinary least squares linear regression of the natural log
transformation of λ and $b_{abs}(\lambda)$. Here, $b_{abs}(\lambda)$ (m$^{-1}$) was calculated by multiplying the
wavelength-dependent, loading artifact-corrected, light-absorbing aerosol concentration reported
by the AE33 by the wavelength-dependent mass absorption cross-section of BC on a filter (m$^2$



g$^{-1}$). Aerosol absorption was calculated per second and then averaged per minute with a 90% data
completeness threshold applied at seven wavelengths measured by the ground aethalometer.

BrC mass concentrations were calculated using the Delta-C method, which estimates BrC

as the difference between UVPM and BC concentrations in units of (μg m$^{-3}$):
$$BrC = UVPM - BC \qquad \text{(Equation 2)}$$
The Delta-C method, as often applied in prior studies, projects BC absorption to the near UV
wavelengths assuming $AAE_{BC} = 1$ and attributes excess light-absorption to BrC, where UVPM
and, thus BrC, are assumed to have the same absorption cross-section as BC (18.47 m$^2$ g$^{-1}$ at 370
nm) (Harrison et al., 2013; Huang et al., 2011; Stampfer et al., 2020; Wagstaff et al., 2022; Wang
et al., 2010, 2011a, b). BrC concentration is thus operationally defined rather than an actual mass
concentration. In this study, we use a BrC absorption cross-section empirically determined by
Ivančič et al. (4.5 m$^2$ g$^{-1}$ at 370 nm) and estimate the contribution of BC to each sample's spectral
attenuation, $b_{abs,BC}(\lambda)$, by attributing all attenuation at 880 nm to BC and extrapolating to other
wavelengths assuming $AAE_{BC} = 0.67$. This is the $AAE_{BC}$ value determined in this study, as
described below in Section 3.3.

Following the approach presented in Kirchstetter and Thatcher (2012), we computed the

contribution of BrC to smoldering smoke aerosol absorption of solar radiation. The contribution
of BrC to spectral absorption in each smoke sample, $b_{abs,BrC}(\lambda)$, is determined by subtracting the
BC absorption from the total absorption:
$$b_{abs,BrC}(\lambda) = b_{abs}(\lambda) - b_{abs,BC}(\lambda) \qquad \text{(Equation 3)}$$
Based on the apportionment of spectral absorption to BC and BrC, we compute the fraction of
spectral radiation for smoldering smoke at each wavelength in the solar spectrum that would be
absorbed by BrC:
$$f_{BrC} = \frac{b_{abs,BrC}(\lambda)}{b_{abs}(\lambda)} \qquad \text{(Equation 4)}$$
Last, we compute the fraction of solar radiation that BrC in the smoldering smoke would absorb
in the atmosphere:
$$F_{Brc} = \frac{\int_{\lambda_1}^{\lambda_2} f_{BrC}(\lambda) \cdot i(\lambda) d\lambda}{\int_{\lambda_1}^{\lambda_2} i(\lambda) d\lambda} \qquad \text{(Equation 5)}$$
where i(λ) is the clear sky air mass one global horizontal  solar spectrum at the earth's surface
(Levinson et al., 2010). We evaluate $F_{BrC}$ using two sets of integration bounds ($\lambda_1$, $\lambda_2$): (1) across



the full solar irradiance spectrum from 300 to 2500 nm that is meaningful for atmospheric
warming and (2) in the near-UV from 300 to 400 nm that is more relevant to tropospheric
photochemistry (Li and Li, 2023; Mok et al., 2016).
The modified combustion efficiency (MCE) is typically used to assess the combustion
completeness during biomass burning and was calculated as the mass fraction of fuel C emitted
as $CO_2$ compared to $CO_2$ and CO (Ward and Radke, 1993):
$$MCE = \frac{\Delta CO_2}{\Delta CO_2 + \Delta CO}$$  (Equation 6)
Background-subtracted concentrations $\Delta CO$ and $\Delta CO_2$ were calculated as the difference between
measured concentrations and background concentrations, the latter of which were established
separately for each of the four days of sampling (listed in Table S2). MCE is unitless, and a value
of 0.9 is commonly used as a threshold to distinguish between flaming-dominated (MCE > 0.9)
and smoldering-dominated (MCE < 0.9) combustion (Selimovic et al., 2018).
Fuel-based BC and BrC emission factors ($EF_i$) in units of grams BC and BrC emitted per
kilogram fuel consumed (g kg$^{-1}$) were calculated by the carbon balance method:(Nelson Jr.,

1982)

$$EF_i = \frac{w_c * V_m}{MW_C} \int_{t_0}^{t_1} \frac{\Delta C_i}{(\Delta CO + \Delta CO_2)} dt$$  (Equation 7)
where $\Delta C_i$ is the background-subtracted BC or BrC concentration (µg m$^{-3}$), $w_c = 0.5$ is the weight
fraction of carbon in the biomass fuel (Urbanski, 2014), $V_m$ is the molar volume of air and equal
to 0.024 m$^3$ mol$^{-1}$, $MW_c$ is the molar mass of carbon and equal to 12 g mol$^{-1}$, and $\Delta CO$ and $\Delta CO_2$
are mixing ratios (ppm). Emission factors were calculated by integration of the background-
subtracted pollutant concentrations over different time intervals. The integration bounds for the
aerial emission factors were the start and end times of each flight, with a temporal basis equal to
the total flight duration, or $t_1 - t_0$ in Eq. 4. Flight durations ranged from 4–22 minutes. For the
ground emission factors, the start time ($t_0$) was when the aethalometer began collecting samples
on a new filter spot and the attenuation (ATN) was zero. The end time ($t_1$) was when the filter
became saturated at an ATN reached 100. At that point, the aethalometer advanced its filter tape.
These integration bounds resulted in a ground sample temporal basis that corresponded to the
ATN cycle of the aethalometer, which ranged from 2–36 minutes. A detailed discussion of the
chosen temporal basis of the emission factors is provided in the SI.




## 3 Results and Discussion

### 3.1 Emission Factors

BC and BrC emission factors measured on the ground and aloft are presented in Table 1.
Overall, the aerial platform measured smoke characterized by a higher modified combustion
efficiency ($MCE_{aerial}$ = 0.88 ± 0.05, average ± standard deviation) and nearly 2 times higher BC
emission factor ($EF_{BC,aerial}$ = 0.92 ± 0.48 g kg$^{-1}$) than the smoke measured on the ground
($MCE_{ground}$ = 0.83 ± 0.03; $EF_{BC,ground}$ = 0.47 ± 0.40 g kg$^{-1}$).

Table 1: Summary Statistics of Carbonaceous Aerosol Emission Factors and MCE (average ±
standard deviation)

|  | Number of Samples | MCE | BC (g kg$^{-1}$) | BrC (g kg$^{-1}$) |
|---|---|---|---|---|
| Aerial samples | 23 | 0.88 ± 0.05 | 0.92 ± 0.48 | – |
| Ground samples | 66 | 0.83 ± 0.03 | 0.47 ± 0.40 | 7.0 ± 2.7 |
| Smoldering samples | 77 | <0.9 | 0.52 ± 0.42 | – |
| Flaming samples | 12 | >0.9 | 1.0 ± 0.48 | – |
| All samples | 89 | 0.84 ± 0.04 | 0.59 ± 0.68 | – |


BC emission factors are plotted against MCE in Figure 1. Individual ground platform
samples are plotted as orange circles and aerial samples are plotted as blue squares. Nearly all
the smoke samples collected from the ground platform (64 of 66 ATN cycles) were associated
with smoldering combustion (MCE < 0.9). A roughly equal number smoke samples collected
aloft were associated with flaming-dominant combustion (10 flights) and smoldering-dominant
combustion (13 flights). BC emission factors demonstrated a weak positive linear correlation
(solid black line, $r^2$ = 0.11) against MCE values, with BC emission factors spanning an order of
magnitude (0.11 to 1.70 g kg$^{-1}$) and MCE values ranging from 0.76 to 0.96. This relationship is
similar to the weak positive linear trend reported by McMeeking et al. (2009) for a laboratory
study ($r^2$ = 0.09), shown as a dashed black line in Figure 1a. In contrast, another laboratory study
by Hosseini et al. (2013) reported a weak negative linear trend (dotted black line, $r^2$ = 0.10). The
application of linear regression models to emission factor data would allow these field and
laboratory studies to be scaled in fuel consumption models as a function of combustion




conditions and/or fire intensity (Burling et al., 2011; May et al., 2014; Ottmar, 2014; Selimovic
et al., 2018; Urbanski, 2014). However, given the very low coefficients of determination from
this work and previous laboratory studies ($r^2 < 0.15$), MCE is not a strong predictor of the BC
emission factor for smoke model estimates.

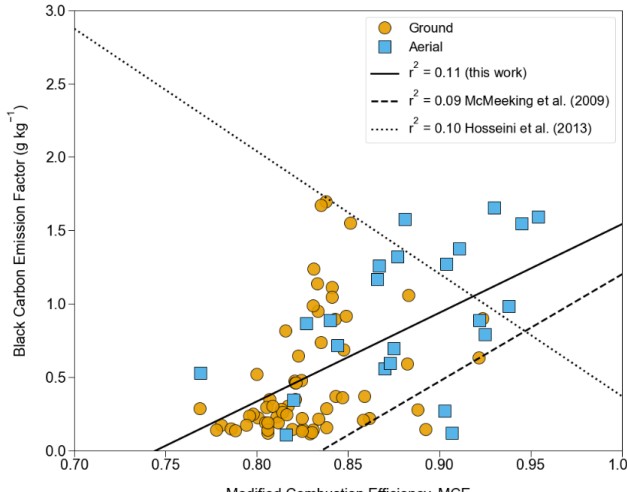


Figure 1: BC emission factors plotted against modified combustion efficiency for all samples.
Symbology designates the ground (circles) and aerial (squares) measurement platforms. All
samples fit with a linear regression model and compared to previous laboratory linear models of
BC emission factors as a function of MCE (Hosseini et al., 2013; McMeeking et al., 2009).

**3.2 Emissions Modeling in Fuel Consumption Models**
BC emission factor distributions for flaming (MCE>0.9) and smoldering (MCE<0.9)
conditions are presented in Figure 2. These combustion categories were chosen to match how
smoke models calculate emissions, often with combustion-phase dependent emission factors.
Fuel consumption models (e.g., Burnup, CONSUME) compute the total fuel consumed
separately during flaming and smoldering combustion phases of a burn. Smoke models then
apply the appropriate EFs, with either one EF for flaming combustion and one EF for smoldering
combustion (e.g., FOFEM), or using a linear model like presented in Figure 1a in which the
calculated MCE in the fuel consumption model is used to obtain the corresponding EF. The
average BC emission factors measured during flaming combustion conditions in this study were
nearly 2 times greater than those measured during smoldering conditions: $EF_{BC,flaming} = 1.0 \pm$



0.48 g kg$^{-1}$ versus EF$_{BC, smoldering}$ = 0.52 ± 0.42, with similar magnitude the average emissions
factors for aerial and ground samples reported above.

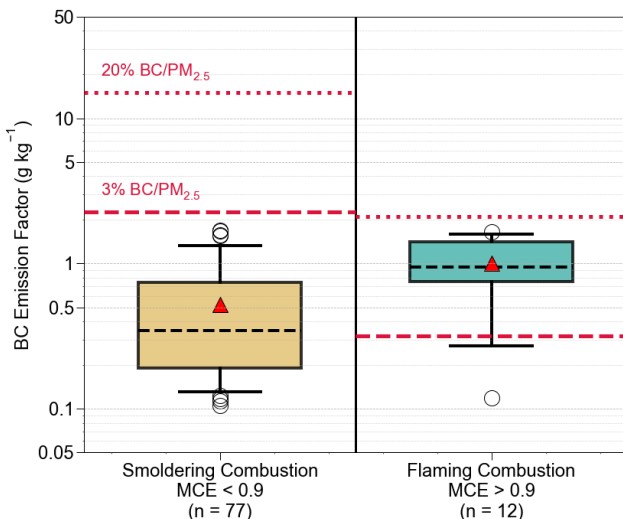


Figure 2: BC emission factors categorized into smoldering combustion (MCE < 0.9) and flaming
combustion (MCE > 0.9) phases for input into fuel consumption. Boxes represent the
interquartile range and tails the 5$^{th}$ and 95$^{th}$ percentile. The median is provided as the black
dashed line across the box, the average as a red triangle, and outliers as open circles. Speciation
profile-derived BC emission factors for 3% and 20% BC/PM$_{2.5}$ for each combustion phase are
plotted as red horizontal dashed and dotted lines, respectively. Note the logarithmic scale on the
y-axis.

Also included in Figure 2 are BC emission factors calculated with the FOFEM

methodology as a fraction of PM$_{2.5}$ emission factors from Burling et al. (2011) for a mixed-
conifer understory prescribed burn (Burling et al., 2011; Lutes, 2020). These BC emission factors
are plotted as horizontal lines across each combustion regime in Figure 2 and represent values
assumed in the most recent federal and California BC inventories. The 2020 National Emissions
Inventory (dashed line) assumes a 3% BC/PM$_{2.5}$ mass ratio for all wildland fire sources,
including prescribed burns and wildfires (US Environmental Protection Agency, 2023). The 2013
California BC Emissions Inventory (dotted line) assumes a 20% BC/PM$_{2.5}$ mass ratio for
prescribed burns (California Air Resources Board, 2016). These BC/PM$_{2.5}$ mass ratios—or BC



speciation profiles—are known to be highly uncertain (Chow et al., 2011). For example, in the
EPA SPECIATE v5.3 database, prescribed burn BC/PM$_{2.5}$ mass ratios vary from 3–11% and for
uncontrolled forest fire or forest fuel types between 0.8–80% (SPECIATE, 2025).
The difference between the average flaming and smoldering BC emission factors
measured in this study and the BC emission factors estimated from BC/PM$_{2.5}$ ratios reveals the
current limitation in using the latter methodology in wildland fire modeling frameworks to
estimate BC emissions. PM$_{2.5}$ emission rates for mixed-conifer forests and many other
ecosystems are higher under smoldering combustion than under flaming combustion, the
opposite of BC emission rates (Burling et al., 2011; Chen et al., 2007). As a result, BC emission
rates are erroneously predicted to be greater under smoldering combustion. The speciation
profiles assumed in the federal and California inventories overestimate BC emission factors
under smoldering combustion for this type of burn by a factor 4 and 29, respectively. Under
flaming combustion, the California inventory overestimates BC emission rates by a factor of 2,
whereas the federal inventory underestimates by 0.3. Dividing the average field BC emission
factors in this study by the literature PM$_{2.5}$ emission factor indicates that a more appropriate BC
speciation profile for a mixed-conifer understory prescribed burn would be 0.7% and 9.5% for
the smoldering and flaming combustion phases, respectively.

**3.3 Optical Properties and Apportionment of Aerosol Solar Radiation Absorption**
BrC emission factors were computed based on ground-level smoke measurements with
the multiwavelength aethalometer, most of which (64 of 66 samples) were during smoldering-
dominated combustion. There as a very weak positive linear relationship ($r^2 = 0.06$) between BrC
emission factors and MCE (Figure 3). The study average BrC emission factor was 7.0 ± 2.7 g
kg$^{-1}$. It is worth noting that this BrC emission factor, computed as described in Section 2.2 based
on an absorption cross-section specific to BrC, is 4.4 times greater than the emission factor
calculated using the more traditional Delta-C method, where the absorption-cross section of BrC
is operationally defined as equal to the absorption cross-section of BC.



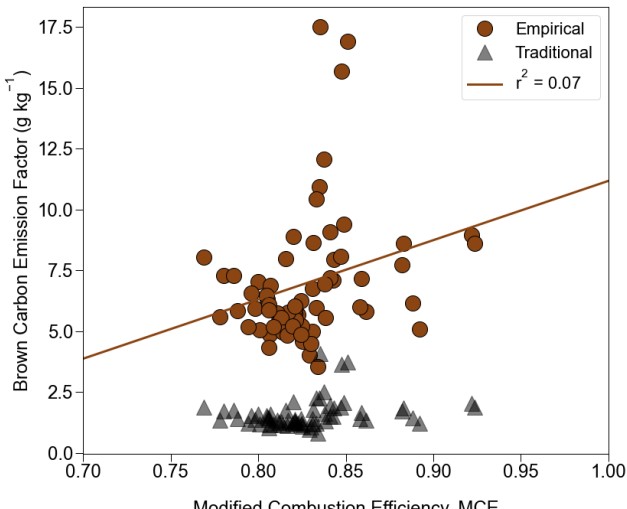


Figure 3: Ground BrC emission factors computed using the Delta-C method with a BrC-specific
mass absorption cross-section (denoted as Empirical and shown with brown circles) and the
more traditional approach using an operationally defined BrC mass absorption cross-section
equal to that of BC (denoted as Traditional and shown with grey triangles) plotted against
modified combustion efficiency. The solid brown line shows the linear regression for the BrC
emission factors calculated with the empirical approach.

Study-average spectral absorption curves are plotted in Figure 4. A power-law fit to the

data over all aethalometer wavelengths from 370–950 nm is shown in Figure 4a. The absorption
data are fit with two trend lines in Figure 4b: an extrapolation of the power law fit to the near-IR
data at 880 and 950 nm to illustrate the BC contribution to total absorption, $b_{abs,BC}(\lambda)$, and a
power law fit of the BrC contribution to absorption, $b_{abs,BrC}(\lambda)$, which extends from mid-visible
wavelengths to the near-UV, calculated using Eq. 3. The AAE given by the power law exponent
reported in Figure 4a is 2.32 (1.35, 3.29; 95% confidence interval), indicating a significant
contribution of BrC to total absorption. The power law fits in Figure 4b yield $AAE_{BrC} = 6.26$
(5.37, 7.13) and $AAE_{BC} = 0.67$. For comparison, El Asmar et al. (2024) found similar overall
AAE = 1.89 (range of 1.31–3.32) and a lower average $AAE_{BrC} = 5.00$ (range of 3.19–7.43) for
prescribed burns in southeastern US measured 0–8 hours downwind with the same model
multiwavelength aethalometer used in this study. The $AAE_{BrC}$ for western wildfires measured
with a photoacoustic spectrometer by Zeng et al. (2022) was also comparable (2.07 ± 1.01;



339 average ± standard deviation). Mie theory predicts that $AAE_{BC} = 1$ for particle diameters less

340 than 10 nm and $AAE_{BC} < 1$ for particle diameters greater than ~0.2 μm (Wang et al., 2016),

341 suggesting that the bulk of sampled aerosols had a diameter greater than 0.2 μm and less than 2.5

342 μm, since a $PM_{2.5}$ cyclone was placed on the sampling inlet.

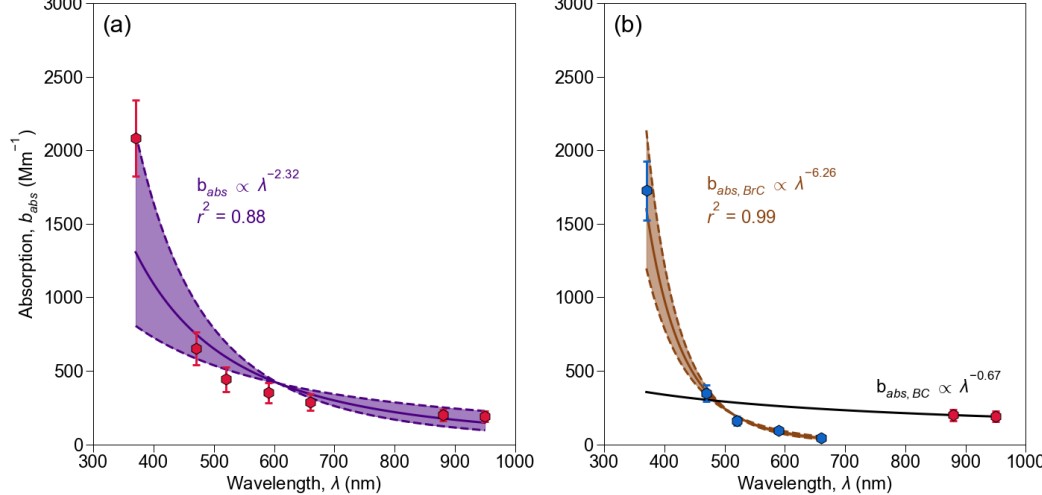

345 Figure 4: Average 1-minute absorption at seven wavelengths measured by the ground

346 aethalometer plotted as red hexagons, with error bars representing 95% confidence intervals. (a)

347 Power-law fit of the average absorption curve at all wavelengths with an AAE = 2.32 and a 95%

348 confidence interval displayed by the shading between the dashed curves. (b) Power-law fit of the

349 BrC average absorption curve (λ = 370, 470, 520, 590, and 660 nm; blue circles) with an $AAE_{BrC}$

350 = 3.43 and a 95% confidence interval displayed by the shading between the dashed curves and

351 the BC average absorptions (λ = 880, 950 nm; red hexagons) with an $AAE_{BC} = 0.67$.

353   Whereas the absorption cross-section of BrC is much lower than that of BC over the near-

354 IR to near-UV portion of the solar spectrum, smoldering smoke emits much more BrC than BC:

355 $7.0 \pm 2.7$ gBrC $kg^{-1}$ versus $0.52 \pm 0.42$ gBC $kg^{-1}$. Consequently, using Equation 5 and shown in

356 Figure S10, we estimate that BrC and BC would account for 23% and 77% of incoming solar

357 radiation absorbed by the smoldering smoke in the atmosphere (integrated from 300 to 2500

358 nm). Furthermore, BrC would contribute 82% of the aerosol absorption of solar radiation at

359 wavelengths below 400 nm and, therefore, may affect tropospheric photochemistry.



AAE values reported in the literature are computed using different approaches. For
example, AAE is commonly derived from data at only two wavelengths and those wavelengths
differ from study to study, which makes direct comparison among studies challenging. To
illustrate this point, we calculated AAE values on 1-minute absorption data from the current
study using three wavelength pairs that approximate prior work. Table 2 reports power law fitting
of (i) 370 and 880 nm to approximate the wavelengths in a photoacoustic extinctiometer, (ii) 470
and 660 nm to approximate a continuous light absorption photometer, and (iii) 470 and 880 nm
to approximate the satellite based AERONET.

Table 2: Measured and Nearest Aethalometer Wavelengths to Calculate the Absorption Ångström
Exponent (AAE)

| Carbonaceous Aerosol Measurement Method | Example Studies | Measured Wavelengths, λ (nm) | Nearest Aethalometer Wavelengths, λ (nm) | AAE, Average ± Standard Deviation |
|---|---|---|---|---|
| Aethalometer (Magee Scientific AE33) | This Work (Butler et al.) El Asmar et al. (2024) | 370, 470, 520, 590, 660, 880, 950 | — | 2.55 ± 0.43 |
| Photoacoustic spectrometer (Droplet Technologies PAX) | Selimovic et al. (2018) Zeng et al. (2022) | 401, 870 | 370, 880 | 2.97 ± 0.54 |
| Continuous light absorption photometer | Marsavin et al. (2023) | 467, 652 | 470, 660 | 2.82 ± 0.59 |
| Satellite (AERONET) | Cazorla et al. (2013) Feng et al. (2013) Wang et al. (2016) Bian et al. (2020) | 440, 870 | 470, 880 | 2.15 ± 0.37 |


The 1-minute average AAE for the three wavelength pairs are listed in the rightmost
column of Table 2. The 370, 880 and 470, 880 wavelength pairs have a 16% and 11% greater
value than the seven-wavelength power law fit in this work, whereas the 440, 870 wavelength
pair a 16% lesser value. These differences in average AAE indicate the uncertainty in interstudy



comparison is approximately ± 15%. Distributions of the coefficient of determination ($r^2$) for
each approach are also presented in Figure S11. A power law fit of 1-minute average data at all
seven wavelengths ($AAE_{7\lambda}$) yielded the highest average coefficient of determination ($r^2 = 0.88$),
followed closely by fitting data at only 370 and 880 nm ($r^2 = 0.87$). The lower average $r^2$ values
for power law fitting of data at 470 and 660 nm ($r^2 = 0.71$) and 470 and 880 nm ($r^2 = 0.60$)
suggest that the AAE values determined from these approaches are not as certain.

**4 Conclusion**

Fuel-based BC and BrC emission factors were calculated by the carbon balance method

with semi-continuous monitoring of a BC, CO, and $CO_2$ on ground and aerial platforms for four
days of prescribed burns. Aerial platform BC emission factors were measured under both flaming
and smoldering combustion, whereas ground platform BC and BrC emission factors skewed
towards almost entirely under smoldering combustion conditions. AAE, an aerosol optical
property, was similarly quantified for smoldering combustion. BC emission factors were found to
be poorly represented by a linear regression model based on MCE and were 2 times higher under
flaming combustion than smoldering combustion. In addition, BC emission factors may be used
in smoke models to improve wildland fire emissions inventories. BrC emission factors, estimated
using a BrC-specific absorption cross-section, were nearly 14 and 7 times greater than the
smoldering and flaming BC emission factors, respectively. The study-average AAE indicated
significant BrC absorption, especially in the near-UV spectrum, indicating that BrC is a
significant contributor to biomass smoke absorption of solar radiation. The $AAE_{BrC}$ reported here
may be parametrized in global earth systems models to represent the contribution of BrC to total
aerosol absorption of incoming shortwave radiation for mixed-conifer prescribed burning (Saleh,

2020).

In future work, deployment of a multiwavelength aethalometer on the aerial platform,

would allow for Delta-C and AAE analyses to estimate BrC emission factors and optical
properties under flaming combustion. Multiwavelength aerosol absorption measurements on an
aerial platform across a wide range of combustion conditions would yield more representative
BrC emission factors and AAE values, which would inform how to model BrC emissions during
different combustion phases in fuel consumption models. Studies that quantify health impacts of





prescribed burn smoke with a chemical transport model will benefit from fuel-based emission
factors in this work and could determine the exposure concentrations of BC and BrC in $PM_{2.5}$.
The overall radiative effects of BC and BrC remains uncertain due to large uncertainties in global
emissions inventories from wildland fires sources (Bond et al., 2013). Further improvements in
bottom-up carbonaceous aerosol emissions inventories would constrain satellite retrievals of
aerosol optical depth used to model aerosol scattering and absorption in global climate models.
To mitigate the health and climate impacts of BC and BrC emissions, prescribed burns
will be critical to promote climate resilient, fire-adapted forest ecosystems. Prescribed burns
consume less fuel per burned area than wildfires by a factor of 2–4, emit less greenhouse gases
and climate pollutants, and have less severe smoke health impacts (Kelp et al., 2023; Kiely et al.,
2024; Ottmar, 2014). Further partnership between government agencies, private land owners,
and tribal nations will likely increase the frequency and effectiveness of prescribed burns (Miller
et al., 2020). Indigenous fire stewardship should be centered in this aim, which uses controlled
fire to change fire regimes in ecosystems to adapt to climate change, encourage certain species
growth, and increase resources to sustain indigenous knowledge, cultural practices, and traditions
(Lake and Christianson, 2019). As prescribed burns increase in prevalence, continued field
measurements of emission factors with state-of-the-science platforms should focus on generating
emission factors for ecosystems and wildland fire activity globally.

**Author Contribution**
**James D.A. Butler:** Conceptualization, Data Curation, Formal Analysis, Investigation,
Methodology, Resources, Visualization, Writing – original draft, Writing – review and editing.
**Afsara Tasnia:** Data Curation, Investigation, Methodology, Resources, Writing – review and
editing.
**Deep Sengupta:** Data Curation, Investigation, Methodology, Resources.
**Nathan Kreisberg:** Data Curation, Investigation, Methodology, Resources, Writing – review
and editing.
**Kelley C. Barsanti:** Conceptualization, Funding Acquisition, Methodology, Project
Administration, Supervision, Writing – review and editing.



**Allen H Goldstein:** Conceptualization, Funding Acquisition, Methodology, Project
Administration, Supervision, Writing – review and editing.
**Chelsea V. Preble:** Conceptualization, Methodology, Resources, Writing – review and editing.
**Rebecca A. Sugrue:** Resources, Writing – review and editing.
**Thomas W. Kirchstetter:** Conceptualization, Funding Acquisition, Methodology, Writing –
original draft, Writing – review and editing.

**Competing interests**
We declare that co-author Barsanti is on the editorial board of the journal *Atmospheric Chemistry*
*and Physics.*

**Code/Data Availability**
Data and code available upon request.

**Acknowledgement**
This work was supported by the California Air Resources Board (CARB) under contract
19RD008 and the California Department of Forestry and Fire Protection (CAL FIRE). Butler,
Kirchstetter, Preble, and Sugrue also acknowledge support of the Department of Energy under
Contract No. DEAC02-05CH11231. The statements and conclusions herein are those of the
authors and do not necessarily reflect the views of the project sponsors. We thank Ariel
Roughton, Rob York, John Battles, Scott Stephens and the staff of the Blodgett Forest Research
Station for their work to conduct the prescribed burns, feed and house the research team, and
ensure safety when taking field measurements; Coty Jen for her feedback on early analyses;
Drew Hill for his assistance on the Delta-C methodology; Adam Wise for development of the
photographs in Key Figure, Figure S2, and Figure S9; and Robert Harley for his review of early
drafts.






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
