# Peer review of "Emission Factors and Optical Properties of Black and Brown Carbon Emitted at a Mixed- Conifer Forest Prescribed Burn"

_EGUsphere, 2025_

## Referee Comment (RC1)

**Emission Factors and Optical Properties of Black and Brown Carbon Emitted at a Mixed-Conifer Forest Prescribed Bun**

The authors presented results from their field measurements of both black carbon (BC) and brown carbon (BrC) emissions from prescribed buns at a mix-conifer forest site in California. The measurement techniques and data analysis were presented succinctly in the manuscript. The main takeaway of the study was also stated clearly that: (1) modified combustion efficiency (MCE) is a poor indicator of BC emission factor; (2) there is a clear difference between the determined BC emission factor and the current modeling inventory; and (3) BrC plays an important role in tropospheric photochemistry and contributes >20% of total solar absorption by smoldering smoke in the atmosphere.

General comment:

The study was described and presented clearly via both equations and figures in both the main manuscript and the supporting information document. The implied significant contribution of brown carbon aerosols to tropospheric photochemistry and total solar absorption in the atmosphere suggests that there should be more field studies of prescribed burns especially at different ecosystems. In addition, the significant difference in BC emission factor between what is currently used in the national and California emission inventory and this study also suggests that we should perform more field measurements to make sure we are representing the fire emissions appropriately in the models.

Specific comments/questions:

L30: the authors should define 'MCE' in the Abstract like they did AAE.

L35: this may not be completely necessary, but it would be nice to include the reference for the delta-C method for readers unfamiliar with it.

L219: "…the filter became saturated at an ATN reached 100." – this part of the sentence reads weird and should be re-written.

L265: There is no Figure 1a. Figure 1?

L 269: similar magnitude "as" the average emissions

L349-350 (Figure 4 caption): should AAE-BrC = 6.26 as mentioned in the text (L333) and on Figure 4b instead of 3.43?

L373-375: did you mean 470,660 wavelength pair as being 11% greater than the 7-wavelength power fit of this work, since the averaged AAE of the former is 2.82 vs. 2.55 of the latter? Similarly, you mean the 470,880 wavelength pair as the 16% lesser value?

If the authors were to conduct additional field measurements of prescribed burns, what ecosystem would they focus on and why? Do the authors anticipate similar BC and BrC emission factors and AAE in a different ecosystem or different region (e.g., North East US)?

---

## Author Response (AR1)

**Reply to Reviewer Comments**

We appreciate the reviewers' time and thoughtful feedback on our manuscript. Their input has helped us improve the quality and clarity of the paper substantially. Below, we address each comment in turn, indicating how we have revised the manuscript in response or clarifying points as needed.

**Reviewer 1**

*Comment 1: L30: the authors should define 'MCE' in the Abstract like they did AAE.*

Response 1: We have replaced 'MCE' with 'Modified combustion efficiency' in the abstract.

*Comment 2: L35: this may not be completely necessary, but it would be nice to include the reference for the delta-C method for readers unfamiliar with it.*

Response 2: We agree with the reviewer that including references to the Delta-C method is essential and have included seven references for the Delta-C method in line 176 Section 2.2, Calculations. We have chosen to forego any references in the abstract.

*Comment 3: L219: "…the filter became saturated at an ATN reached 100." – this part of the sentence reads weird and should be re-written.*

Response 3: We have rewritten this sentence for grammar, it now reads: "The end time ($t_1$) was when the filter became saturated at an ATN of 100."

*Comment 4: L265: There is no Figure 1a. Figure 1?*

Response 4: We thank the reviewer for catching these typos and have deleted references to Figure 1a.

*Comment 5: L 269: similar magnitude "as" the average emissions L349-350*

Response 5: We have added the word "as" where the reviewer suggests.

*Comment 6: (Figure 4 caption): should AAE-BrC = 6.26 as mentioned in the text (L333) and on Figure 4b instead of 3.43?*

Response 6: Yes, this is an error in the figure caption and has been updated as the reviewer suggests.

*Comment 7: L373-375: did you mean 470,660 wavelength pair as being 11% greater than the 7-wavelength power fit of this work, since the averaged AAE of the former is 2.82 vs. 2.55 of the latter? Similarly, you mean the 470,880 wavelength pair as the 16% lesser value?*

Response 7: Yes, we mean that the 470,660 wavelength pair was 11% greater and the 470,880 wavelength pair was 16% lesser than the study average AAE value. This statement was updated to reflect the correct percentage differences in AAE values:

> The 1-minute average AAE for the three wavelength pairs are listed in the rightmost column of Table 2. The 370, 880 and 470, 660 wavelength pairs have a 16% and 11% greater value than the seven-wavelength power law fit in this work, whereas the 470, 880 wavelength pair have a 16% lesser value.

*Comment 8: If the authors were to conduct additional field measurements of prescribed burns, what ecosystem would they focus on and why? Do the authors anticipate similar BC and BrC emission factors and AAE in a different ecosystem or different region (e.g., North East US)?*

Response 8: We appreciate this question from the reviewer and have added text to the conclusion in lines 434–440:

> Continued field measurements of emission factors with state-of-the-science platforms should focus on characterizing emission factors and optical properties for ecosystems commonly burned in the western United States, like the mixed-conifer forests studied here, Ponderosa pine forests, coastal forests, chaparral shrublands, and oak savannas. Carbonaceous aerosol emission factors for each of these ecosystems remain understudied, especially for BrC, and likely vary across ecosystems depending on fuel moisture content, fuel types, and combustion efficiency of burn.

**Reviewer 2**

*Comment 1: Describe in more detail how representative sampling with the drone was done given the issue with downwash from the props. Was the sampling isokinetic, etc?*

Response 1: The reviewer raises an important concern regarding the representativeness of drone sampling, which may not completely capture the plume, as the drone sampling captures a single point above an area source emission. For vertical profiling, where concentration as a function of altitude is important, downwash can be a serious confounding problem. However, for this study, we were not trying to measure vertically resolved concentrations. Rather, prescribed fire smoke was sampled in heavy plumes without regard to sampling column height and

integrated over entire flights, similar to previous emission studies like Aurell et al. (2023). Further, isokinetic sampling is not important for sub-micron particles that remain well attached to the sampling airstream. The authors, therefore, believe losses of particles to drone interference or non-isokinetic sampling were minimal. The following statement was added to the manuscript at line 135: "The aerial platform was flown in the densest smoke plumes to intercept the bulk of the prescribed burn emissions and hovered within these plumes to capture fresh emissions representative of the event."

*Comment 2: Briefly discuss the limitations of the AE33.*

Response 2: We thank the reviewer for noting that we did not explicitly direct the reader to the Supporting Information where the limitations of the AE33 are discussed in detail. To clarify we have added a statement at line 150 to direct the reader there:

> Filter-based aerosol absorption photometry has well known limitations due to the interactions of the collected aerosol particles and filter media. Corrections for these sampling artifacts are detailed in the Supporting Information, SI.

*Comment 3: Regarding the Delta-C approach discussed in the Methods. The method seems flawed, and it is not clear from this discussion why it is utilized. The instrument (AE33) used at the ground site measures light absorption coefficients. The instruments may convert this light absorption measurement and report a mass concentration with an appropriate MAC. The MACs used for this conversion are known so the bap at the 7 wavelengths is known. For the delta-C, specifically state where the numbers in Eq2 come from, how does the AAE of BC = 1 come into the calculation. Where does the assumed MACs for BrC and BC being equal come from. State why exactly this method is used, since a different approach based on Eq3 is also used. Overall, should one continue to promote a flawed analysis?*

Response 3: Thank you for pointing out that this section of the manuscript was confusing. We agree and have revised this section to clarify how we calculate BrC concentrations.

The Delta-C method (Equation 2) attributes excess UV light absorption—beyond that absorbed by BC—to BrC. However, the AE33 aethalometer reports UVPM concentrations based on the assumption that UVPM, and thus BrC, has the same mass absorption cross-section as BC. This is an outdated assumption. It is now well established that the mass absorption cross-section of BrC differs substantially from that of BC. In fact, the newest aethalometer model (AE36) uses the BrC absorption cross-section reported in Ivančič et al. (2022).

As now clarified in the revised text on lines 185–189, we first apportion the total light absorption at 370 nm between BrC and BC (Equation 3). Consistent with the approach used in the newest aethalometer, we then divide BrC light absorption by the best-estimate of the BrC mass absorption cross-section to derive the BrC concentration.

In comment 6, the reviewer questions our approach to apportioning light absorption at 370 nm. While a common approach is to assume $AAE_{BC} = 1$, we chose $AAE_{BC} = 0.67$. The former value is consistent with Mie theory and strictly applies to uncoated, spherical particles. The latter value of 0.67 was estimated based on the measured absorption in the near-IR (at 880 nm and 950 nm). We agree with the reviewer that extrapolating to the near-UV from two near-IR wavelengths induces some level of uncertainty. We chose this approach because it is based on measurements of the smoke sampled in this study and that seemed better than using the theoretical value for uncoated spherical particles. The value of 0.67 is within the range of values for BC AAE (Liu et al., 2018, Wu et al. 2024).

The following revised text has been updated on lines 189–197 to clarify this section:

> Whereas $AAE_{BC} = 1$ is a commonly used value that is consistent with Mie theory (i.e., uncoated, ideal spherical particles with wavelength-independent refractive index) (Liu et al., 2018), the value 0.67 was estimated based on the optical properties of the smoke measured in this study, as described below in Section 3.3. We then calculate BrC concentration by dividing $b_{abs,BrC}$(370nm) by the current best estimate of the BrC mass absorption cross-section empirically determined by Ivančič et al. (4.5 $m^2$ $g^{-1}$ at 370 nm) rather than assuming BrC and BC have the same absorption cross-section. In doing so, our BrC concentrations are equivalent to those reported by the newest model of the aethalometer (i.e., the AE36) (Aerosol d.o.o., 2024).

*Comment 4: Line 203. Why is background CO = 0 (Table S2). This is not possible. What is the variability in the background CO2 (this is not given in Table S2), how does this affect the uncertainty of the MCE?*

Response 4: We agree that background CO cannot equal zero and have removed these columns from Table S2. In the SI text, lines 117–121, we have added text describing our reasoning for calculating excess CO assuming a background concentration of zero:

> For CO, the background concentration was assumed to be zero, as no other sources of incomplete combustion were present at the burn and measured CO concentrations were 1–2 orders of magnitude larger than a trace atmospheric background concentration of around 0.3 ppm. Excess $CO_2$ was calculated after subtracting the background concentrations listed in Table S2.

*Comment 5: Calculation of emissions using Equation 7. This is a highly uncertain calculation, and the limitations should be discussed. For example, what is the uncertainty in the BrC concentration which depends on the MAC used, which in turn depends on the type of BrC (see Saleh et al), which can change throughout the burning process. What is the uncertainty in weight fraction of carbon in the fuel (wc). What are the assumptions on which this formula is*

*based. Eg, does it assume that by mass most C emitted is in the form of CO and CO2 – what about other forms of carbon, eg, OA? How does ground sampling at one location (over a short period of time), provide the emission factor representative of a large area with different fuels and properties (moisture content). The implicit assumption here (see statements made in the Conclusions) is that these reported EF apply broadly to these types of forests/fuels. Discuss the many assumptions and limitations.*

Response 5: Equation (7) is broadly accepted in wildland fire literature as the standard equation to calculate an emission factor, such as in Aurell and Gullet (2013) and Ward and Radke (1993). Andreae (2019), Akagi et al. (2011), and Binte Shahid et al. (2024) report emission factor datasets where CO and CO2 comprise 90–98% of total carbon emitted, with a small fraction emitted as organic gases (~2%) and aerosols (~2%). In Yokelson et al. (2013), CO and CO2 were 94% of total carbon emitted from a forest understory prescribed fire. A review of forest carbon storage and IPCC methodology reported the average wood C content as 50.8% ± 0.7% (95% CI) for conifer fuels (Thomas and Martin, 2012). We have added this reference when describing the weight fraction of carbon in line 226. Prescribed fires are dynamic events that evolve throughout sampling. We acknowledge that further studies are required to constrain this uncertainty for a western US mixed-conifer understory prescribed burns.

With respect to the representativeness of emission factors in this work, we agree that this is important and direct the reviewer to the discussion in the Supporting Information, Temporal Basis of Emission Factors section. This section has been renamed to "Representativeness and Temporal Basis of Emission Factors". This discussion includes an analysis of the distribution of emission factors depending on the temporal basis used in Equation (7).

*Comment 6: Fig 4, specify what the error bars represent (eg, Aeth flow uncertainty…?), and how are they determined. Or was this data variability? Why in Fig 4a does the 95% CI converge to zero at about 600 nm. Fig 4b, BC AAE was determined based on two wavelengths? Seem highly uncertain (this is noted in a different context later in the paper). Justify that based on the extrapolation from 2 data points (wavelengths), the inference of BC size is reasonable, ie lines 339-342.*

Response 6: The error bars on the red hexagons in Figure 4 represent a 95% confidence interval in the minutely absorption values across all sampling in the study. The bounding absorption curves represent the 95% confidence interval values for AAE, which are the shallowest and steepest possible absorption curves (power-law fits) in linear space. AAE was calculated as an ordinary linear regression in log-log space on the log-transformed absorption data and measured wavelengths. The conversion of AAE from log-log space to linear space causes the intersection of the bounding 95% confidence interval curves. The uncertainty in absorption around 600 nm should be interpreted through the red error bars on the point at in Figure 4.

This text was clarified in the Figure 4 caption starting in line 365:

(a) Power-law fit of the average absorption curve at all wavelengths with an AAE = 2.32 (solid curve) and 95% confidence interval AAE values displayed as the bounding dashed curves. (b) Power-law fit of the BrC average absorption curve (l = 370, 470, 520, 590, and 660 nm; blue circles) with an $AAE_{BrC}$ = 6.26 (solid brown curve) with 95% confidence interval AAE values displayed as the bounding dashed curves and the BC average absorptions (l = 880, 950 nm; red hexagons) with an $AAE_{BC}$ = 0.67 (solid black curve).

The uncertainty associated with the BC AAE is discussed above in our response to Comment 3.

*Comment 7: Why is the term Fuel-based emission factors used? Are the fuels and its conditions known? They don't seem to be reported in any detail, only just general forest type is given.*

Response 7: The term "fuel-based" is used to indicate that the emission factors were computed per kilogram of fuel consumed, as opposed to per acre burned, which would require that the fuel loading of the burn plot be known. Fuel-based emission factors are standard inputs to fuel consumption models like Burnup and CONSUME. The carbon balance method is very commonly applied in biomass smoke emission studies and it is well established that CO and CO2 typically account for 90–98% of total emitted carbon (see Response 5). We have added text to reference these studies in line 228:

> In Equation (7), the carbon balance method assumes that all fuel carbon is emitted as either CO or CO2, given 90–98% of total emitted carbon is emitted as these gases (Akagi et al., 2011; Binte Shahid et al., 2024; Nelson Jr., 1982; Yokelson et al., 2013).

*Comment 8: Line 393-394 is not clear. What does the respectively refer to?*

Response 8: We have rewritten this statement on lines 413–415 to make it clear that smoldering BrC emission factors were nearly 14 times greater than smoldering BC emission factors and 7 times greater than flaming BC emission factors:

> BrC emission factors, estimated using a BrC-specific absorption cross-section, were nearly 14 times greater than smoldering BC emission factors and 7 times greater flaming BC emission factors.

*Comment 9: Line 397 to 399. This is an odd statement. Anything MAY be parameterized and used in global perditions, but it doesn't mean it is correct. It is highly speculative to suggest that the very limited data reported here can be applied for global predictions.*

Response 9: We agree with the reviewer it would be impractical to parameterize the AAE for brown carbon from this one study into a global earth system model and have deleted the sentence.

*Comment 10: Line 405-407 discuss use of this data to predict adverse health impacts. This is highly speculative and not supported by this study. For one, it assumes all BrC, as determined by this method, has a similar toxicity.*

Response 10: Similar to Comment 9, we agree with the reviewer that this study does not investigate the health impacts of BC or BrC and we struck this paragraph. We have instead added text in lines 440–442 to emphasize how future studies could investigate the relative toxicity BC and BrC emitted by prescribed burns:

> In parallel, future studies could also investigate the toxicity of BC and BrC emitted by prescribed burns, which may vary depending on combustion conditions and fuels burned.

*Comment 11: Does this work consider dark BrC or tar balls. Where do they contribute, ie, to BrC or BC? How do they effect the calculations of AAEs since the method used assumes absorption at the two highest wavelengths are only from BC, but tar balls (or dark BrC ) also absorb light at these high wavelengths.*

Response 11: We did not measure tar balls in this work; however, we added a statement that tar balls may contribute aerosol absorption for BrC in line 417:

> A fraction of this BrC absorption may be attributable to so-called tar balls, which may comprise 5–30% of total $PM_{2.5}$ in wildfire smoke in the western United States (Adachi et al., 2024; Chakrabarty et al., 2023).

*Comment 12: Line 415-416. Doesn't the health impact of prescribed vs wildfires depend on location (extent of exposures), or is PM toxicity being discussed here? This is possibly too broad a statement. Essentially, don't all the statements here come down to simply that wildfires consume much more fuel than prescribed fires so have greater emissions and hence possible impacts.*

Response 12: We agree with the reviewer that in general wildfires consume more fuel, resulting in higher emissions and consequent health impacts, as the referenced studies demonstrate. We have deleted this sentence and added a concluding statement in line 417 about how future work could investigate the toxicity of BC and BrC emitted from prescribed burns, as noted in Response 10.

*Comment 13: Line 418-421 regarding "Indigenous fire stewardship should be centered in this aim". This is an opinion and not a conclusion based on this study. There may be benefits to include indigenous knowledge, but why must it be the main focus (ie, centered)?*

Response 13: We thank the reviewer for their comment and deleted this sentence from the conclusion.

*Comment 14: It is not clear how one can discuss BC/PM2.5 mass emission factors since PM2.5 mass was not measured (or was it)?*

Response 14: While PM2.5 mass emission factors were not directly measured, BC/PM2.5 emissions ratios were calculated with PM2.5 emission factors from Burling et. al (2011), which also sampled the mixed-conifer forest fuels, the same ecosystem type as was sampled in this study. The purpose of discussing BC/PM2.5 mass emission factors was to demonstrate the high uncertainty in using emissions ratios in emission inventories and suggest direct measurement of BC emission factors is preferred. We have updated the statement about BC/PM2.5 emissions ratios to include explicitly reference Burling et al. (2011) in line 322:

> Dividing the average field BC emission factors in this study by the $PM_{2.5}$ emission factor from Burling et al. (2011) indicates that a more appropriate BC speciation profile for a mixed-conifer understory prescribed burn would be 0.7% and 9.5% for the smoldering and flaming combustion phases, respectively.

*Comment 15: Many studies have assessed the radiative importance of BrC relative to BC similar to what was done here. It would be useful to add comparisons to other studies to put these results in context.*

Response 15: We have added the following statement to compare our BrC aerosol absorption result to a recent study of wildfire smoke in line :

> Similarly, Chakrabarty et al. (2023) found BrC contributes 66–86% of total aerosol absorption at 405 nm in wildfire smoke emitted in the western United States.